# Altered central pain processing in fibromyalgia—A multimodal neuroimaging case-control study using arterial spin labelling

Monika Müller [1,2], Florian Wüthrich [2], Andrea Federspiel [2], Roland Wiest [3], Niklaus Egloff [4], Stephan Reichenbach [5,6], Aristomenis Exadaktylos [7], Peter Jüni [8,9], Michele Curatolo [10]*, Sebastian Walther [2]

1 University Clinic of Anesthesiology and Pain Medicine, Inselspital, Bern, Switzerland, 2 Translational Research Center, University Hospital of Psychiatry and Psychotherapy, Bern, Switzerland, 3 Department of Neuroradiology, University Clinic of Radiology, Inselspital, Bern, Switzerland, 4 Department of Psychosomatic Medicine, University Clinic of Internal Medicine, Inselspital, Bern, Switzerland, 5 University Clinic of Rheumatology, Clinical Immunology and Allergology, Inselspital, Bern, Switzerland, 6 Institute of Social and Preventive Medicine, University of Bern, Bern, Switzerland, 7 Emergency Services of the University Hospital of Bern, Inselspital, Bern, Switzerland, 8 Applied Health Research Centre (AHRC), Li Ka Shing Knowledge Institute of St. Michael's Hospital, Department of Medicine, University of Toronto, Toronto, Canada, 9 Institute of Health Policy, Management and Evaluation, University of Toronto, Toronto, Canada, 10 Department of Anesthesiology and Pain Medicine, University of Washington, Seattle, Washington, United States of America

* curatolo@uw.edu

**Data Availability Statement:** All relevant data are within the manuscript and its Supporting information files.

## Abstract

Fibromyalgia is characterized by chronic pain and a striking discrepancy between objective signs of tissue damage and severity of pain. Function and structural alterations in brain areas involved in pain processing may explain this feature. Previous case-control studies in fibromyalgia focused on acute pain processing using experimentally-evoked pain paradigms. Yet, these studies do not allow conclusions about chronic, stimulus-independent pain. Resting-state cerebral blood flow (rsCBF) acquired by arterial spin labelling (ASL) may be a more accurate marker for chronic pain. The objective was to integrate four different functional and structural neuroimaging markers to evaluate the neural correlate of chronic, stimulus-independent pain using a resting-state paradigm. In line with the pathophysiological concept of enhanced central pain processing we hypothesized that rsCBF is increased in fibromyalgia in areas involved in processing of acute pain. We performed an age matched case-control study of 32 female fibromyalgia patients and 32 pain-free controls and calculated group differences in rsCBF, resting state functional connectivity, grey matter volume and cortical thickness using whole-brain and region of interest analyses. We adjusted all analyses for depression and anxiety. As centrally acting drugs are likely to interfere with neuroimaging markers, we performed a subgroup analysis limited to patients not taking such drugs. We found no differences between cases and controls in rsCBF of the thalamus, the basal ganglia, the insula, the somatosensory cortex, the prefrontal cortex, the anterior cingulum and supplementary motor area as brain areas previously identified to be involved in acute processing in fibromyalgia. The results remained robust across all neuroimaging markers and when limiting the study population to patients not taking centrally acting drugs

**Funding:** The research team with Michele Curatolo as main applicant were awarded a grant by the Bangerter-Rhyner Foundation to support the present study. The funders had no role in study design, data collection and analysis, decision to publish, or preparation of the manuscript.

**Competing interests:** The authors have declared that no competing interests exist.

and matched controls. In conclusion, we found no evidence for functional or structural alterations in brain areas involved in acute pain processing in fibromyalgia that could reflect neural correlates of chronic stimulus-independent pain.

## Introduction

Fibromyalgia is characterized by chronic and widespread pain with additional symptoms such as fatigue, sleep disturbance and cognitive dysfunctions [1–3]. The Impact on quality of life is comparable to other chronic diseases such as rheumatoid arthritis, diabetes mellitus and chronic obstructive lung disease [4, 5]. Therapeutic options remain limited with modest effects for most treatments and a high proportion of patients not responding to any treatment [6]. Despite the clinical significance of the disease, pathophysiological processes remain poorly understood which limits the development of diagnostic markers and novel treatments targeting pathophysiological mechanisms rather than disease symptoms.

The striking discrepancy between objective signs of tissue damage and magnitude of pain suggests a pathophysiological process involving the central nervous system with possible alterations in brain function and structure [7, 8]. A state of enhanced central pain response associated with increased neural activity in pain processing brain areas may lead to exaggerated pain response even to non-painful stimuli, high stimulus-independent pain and widespread pain [7, 9–14]. Brain areas identified to be involved in pain processing include subcortical regions such as thalamus and basal ganglia and the insula, somatosensory cortex, prefrontal cortex, anterior cingulum and supplementary motor area as cortical regions [8, 15, 16]. It remains however a subject of debate whether these brain areas are mainly processing acute pain signals or whether they also involved in chronic pain [8, 17]. Potential neuroimaging markers of chronic pain include resting-state cerebral blood flow (rsCBF) or functional connectivity on a functional level, as well as alterations of grey matter volume and cortical thickness on a structural level.

First neuroimaging studies investigating functional markers mainly focused on the comparison of acute pain processing mechanisms in fibromyalgia patients and pain-free controls using experimentally-evoked pain paradigms and blood oxygenation level dependent (BOLD) contrasts [7, 8]. In line with the pathophysiological concept of enhanced central pain response, these studies detected increased neural activity in the amygdala, the insula, the somatosensory cortex and the cingulate cortex of fibromyalgia patients after painful stimuli. However, for two reasons, these studies do not allow conclusions about processing of chronic, stimulus-independent pain that typically remains constant over time [18–20]. First, the studies used experimentally evoked acute pain paradigms rather than resting-state paradigms that are more likely to capture neural adaptation to chronic pain. Second, even if the researchers acquired resting-state data, they used BOLD contrasts and quantified resting-state connectivity as marker of chronic pain [21, 22] rather than specific regional resting-state neural activity. However, while BOLD provides a relative and indirect measure of brain activity, rCBF gives an absolute value that is directly related to local brain metabolism and thus appears perfectly suited to quantify neural activity in chronic pain.

Arterial spin labelling (ASL) as an advanced neuroimaging method quantifies resting-state cerebral flow (rsCBF) as a direct marker of neural activity at rest [17, 23]. First studies using ASL in different chronic pain conditions such as chronic low back pain [24, 25], postsurgical pain [26], trigeminal neuropathy [27], postherpetic neuralgia [28] and fibromyalgia [29] are

emerging. Although these studies employed a resting-state paradigm, they only partly accounted for important confounders such as depression, age, and gender, while they failed to consider centrally acting drugs potentially biasing neuroimaging markers. This may explain why the results of the studies remain inconclusive. Additionally, previous studies did not integrate functional and structural neuroimaging by applying multimodal neuroimaging.

We conducted a multimodal neuroimaging study integrating functional and structural markers to explore the neural correlates of chronic pain in fibromyalgia. The primary objective was to compare rsCBF patterns of fibromyalgia patients and pain-free controls using ASL. In line with the paradigm of enhanced central pain response we hypothesized that rsCBF in areas involved in pain processing is increased in fibromyalgia. The secondary objective was to compare resting-state functional connectivity between pre-specified Regions of Interest, grey matter volume and cortical thickness between cases and controls assuming a reduction in these neuroimaging markers in fibromyalgia. We adjusted all our analyses for depression and anxiety and conducted a subgroup analysis limited to patients free of any centrally acting drugs.

## Material and methods

### Participants and study design

We performed a 1:1 frequency age-matched case-control study in fibromyalgia patients and pain-free controls using 10 years age bands. We centrally recruited right-handed [30] female participants at the University Hospital of Bern, Switzerland. We randomly sampled cases from a pool of 238 patients who were first diagnosed with fibromyalgia at either the Department of Rheumatology, the Department of Psychosomatic Medicine or the Pain Clinic between November 2013 and January 2015. We randomly sampled controls from a pool of 9253 women who presented during the same period at the Emergency Department. We stopped sampling when we reached the required sample size. We included controls if they did not suffer from any chronic pain disorder and were pain-free two weeks prior to neuroimaging. We included cases if they were confirmed with the diagnosis of fibromyalgia according to the diagnostic criteria of the American College of Rheumatology at repeat clinical examination at enrollment [2]. Their diagnostic criteria are based on the Widespread Pain Index, which characterizes spread of pain, and the Symptom Severity Scale, which assesses different core symptoms of fibromyalgia including depression that is frequently encountered in these patients. We therefore did not exclude patients with co-morbid depression from the study but adjusted our analyses for severity of depression measured by the Beck-Depression Inventory version 2 [31] as described in the statistical section.

Common exclusion criteria for cases and controls were conditions interfering with the MRI acquisition; neurologic co-morbidity or history of neurosurgical intervention; psychiatric co-morbidity other than unipolar depressive disorder; end-stage somatic co-morbidity defined as untreatable disease that represents a threaten for life in the next few months; intake of strong opioids or any psychopharmacological treatment other than anticonvulsants or antidepressants; inability to understand the consequences of study participation; and pregnancy. We performed the study according to a prospective protocol approved by the ethics committee of the Canton Bern, Switzerland (KEK 43/13), and in accordance with the Declaration of Helsinki [32]. All participants gave written informed consent.

### Assessment of socio-demographic, psychological and clinical characteristics

We assessed the following socio-demographic and psychological characteristics in all participants: age; education (higher education vs lower education); marital status (married vs not married); depression and anxiety. We considered participants with high school or university

degree as having higher education. We used the Beck Depression Inventory version 2 (BDI-II) [31] and the State-Trait-Anxiety-Inventory (STAI) [33] to assess depression and anxiety in all participants. In cases we additionally assessed pain intensity, pain duration, spread of pain and disability. We used the Numerical Rating Scale to measure average pain intensity within the last 24 hours, pain intensity immediately before neuroimaging (NRS, 0 = no pain to 10 = worst pain) and the Fibromyalgia Impact Questionnaire to assess disability (FIQ, 0 = no disability, 100 = most severe disability) [34]. We used the Widespread Pain Index to characterize the spread of pain with (WPI, 0 = no body region with pain to 19 = generalized pain affecting all body regions) [2]. We recorded long-term daily intake of centrally acting drugs such as light opioids, antidepressants or anticonvulsants. The patients took these drugs also on the day of neuroimaging.

## Neuroimaging

**Image acquisition.** We performed multimodal neuroimaging at the Institute of Neuroradiology of the University Hospital of Bern to acquire four neuroimaging markers: resting-state cerebral blood flow (rsCBF); resting-state functional connectivity (rsFC); grey matter volume (GMV); and cortical thickness (CoTh). We performed the MRI with a 3-Tesla Trio whole-body scanner using a 12-channel radio-frequency head coil (Siemens Medical, Germany). All study participants were instructed to lie quietly with eyes closed not thinking of anything particular during functional scans. We obtained the sequences in the following order:

First, high resolution anatomical T1* weighted images: 176 sagittal slices with 256 × 224 matrix points with a non-cubic field of view (FOV) of 256 mm × 224 mm, yielding a nominal isotropic resolution of 1 mm$^3$ (i.e. 1mm × 1mm × 1mm), repetition time (TR) = 7.92ms, echo time (TE) = 2.48ms, flip angle = 16˚, inversion with symmetric timing (inversion time 910ms).

Second, a set of 80 functional T2* weighted images using a pseudo-continuous arterial spin labelling sequence (pCASL) which corresponds to 40 pairs of unlabeled and labeled images [35, 36]: eighteen axial slices at a distance of 1.0 mm; slice thickness = 6.0 mm; FOV = 230 x 230 mm$^2$; matrix size = 128 x 128, yielding a voxel-size of 1.8mm × 1.8mm × 6mm; TR = 4000ms; TE = 18ms. The gap between the labeling slab and the proximal slice was 90 mm; gradient-echo; echo-planar readout; ascending order; acquisition time 45 ms per slice. Slice-selective gradient 6 mT/m, post-labeling delay $w$ = 1250 ms, tagging duration $\tau$ = 1600 ms.

Third, BOLD functional T2* weighted images were acquired with an echo planar imaging (EPI) sequence: 32 axial slices, FOV = 192x192 mm$^2$, matrix size = 64x64, gap thickness = 0.75 mm, resulting in a voxel size of 3x3x3 mm$^3$, TR/TE 1980ms/30ms, flip angle = 90˚, bandwidth = 2232Hz/Px, echo spacing = 0.51ms, 460 volumes.

**Selection of Region of Interests (ROIs).** Even though our main statistical approach was to perform voxel- and vertex-wise whole-brain analyses, we also investigated several pre-specified cortical Regions of Interests (ROIs) likely to be involved in pain processing in fibromyalgia based on previous evidence for functional neuroimaging markers [8] and on atlases for structural markers [37, 38]. We did not investigate software specific ROIs. For functional analyses of rsCBF and rsFC we defined the exact coordinates of 11 ROIs according to a recent meta-analysis by Dehghan et al [8]: Bilateral insula, left anterior and middle cingulate cortex (ACC, MCC), right Amygdala, bilateral superior temporal gyrus (STG), right lingual gyrus and left primary and secondary cortex (S I, S II). Notably, the right insula is represented in two ROIs in this selection while all other brain regions are represented by one ROI each. We created a box around the coordinates with size = 27 voxels (216 mm$^3$). For structural analyses of GMV and CoTh we selected brain regions that represent the above specified regions from existing

atlases. This resulted in 10 volumetric ROIs based on the Automatic Anatomical Labeling (AAL)-Atlas for GMV analyses [37] and 23 ROIs based on the Destrieux-Atlas which is based on gyral and sulcal surface for CoTh analyses [38]. We extracted rsCBF and grey matter values in for ROIs using MarsBaR (Version 0.44, http://marsbar.sourceforge.net/).

**Image preprocessing and calculation of functional and structural neuroimaging markers.**   We performed structural preprocessing for rsCBF, and GMV analyses in SPM12 (Revision 6225, Welcome Trust, London, U.K., https://www.fil.ion.ucl.ac.uk/spm/). T1* images were segmented into gray matter, white matter and cerebrospinal fluid, normalized to the Montreal Neurological Institute (MNI) space and smoothed using an 8 mm full-width at half maximum (FWHM) Gaussian kernel. To compute GMV, defined as relative proportion of grey matter in a region, accounting for brain size and shape, we applied voxel-based morphometry in SPM12 to the preprocessed T1* images applying an absolute threshold of 0.2 to the modulated grey matter segmentations. We calculated the total intracranial volume by adding the segmented grey matter, white matter and cerebrospinal fluid volumes. We carried out preprocessing for CoTh analyses using the standard FreeSurfer pipeline for cortical reconstruction 5.3.0 (Laboratory for Computational Neuroimaging at the Athinoula A. Martinos Center for Biomedical Imaging, Boston, U.S.A., http://surfer.nmr.mgh.harvard.edu/). This includedautomatic motion correction, segmentation, intensity normalization, inflation and registration to a spherical atlas.

We quantified resting-state cerebral blood flow (rsCBF, ml/100g/min) according to a previously applied, standardized protocol [39–41] using the following formula:

$$CBF = \left( \frac{\lambda \cdot \Delta M}{2 \cdot \alpha \cdot M_0 \cdot T_{1b}} \right) \cdot \left( \frac{1}{e^{-w/T_{1b}} - e^{-(\tau+w)/T_{1b}}} \right),$$

where $\Delta M$ is the difference between labeled and control image; $\lambda$ the blood/tissue water partition coefficient (assumed 0.9); $\alpha$ the tagging efficiency (assumed 0.85); $M_0$ the equilibrium brain tissue magnetization; $\tau$ the tagging duration; $w$ the post-labeling delay; and $T_{1b}$ the longitudinal relaxation time of blood (1650ms) [40]. This resulted in 40 unprocessed rsCBF maps per subject. They were realigned and co-registered to the corresponding raw T1* image and normalized using the deformation matrix of the corresponding T1* image. We smoothed the resulting images with a 8mm FWHM Gaussian kernel. We then calculated grey matter rsCBF applying grey matter masks based on the segmented grey matter T1* images which were thresholded at 0.3. We conducted subject-wise first-level generalized linear models with rsCBF of the grey matter as outcome variable and the rsCBF of the white matter, the rsCBF of the cerebrospinal fluid and the realignment parameters as explanatory variables to correct for residual motion and artifacts. We checked six motion parameters (x-, y-, z-translations, roll, pitch, and yaw) and set a limit of two voxels of motion for exclusion. This resulted in 40 preprocessed grey matter rsCBF maps per subject. There was no significant difference between groups in motion and we did not exclude any subject due to excessive motion. To increase the signal-to-noise ratio, we finally modelled a mean rsCBF map for each subject based on the 40 preprocessed maps. We used this mean rsCBF map for further analysis and computed mean global rsCBF within grey matter per subject based on this mean rsCBF map. Preprocessing of rsCBF maps was performed in SPM12 and Matlab (MATLAB 2015a; The MathWorks, Inc., Natick, MA, USA).

BOLD images were co-registered to the corresponding raw T1* image and then processed using the standard processing pipeline in CONN (Version 15, http://www.nitrc.org/projects/conn). This included realignment; slice-time correction; outlier-detection using ART-toolbox (global-signal z threshold 9, subject motion threshold 2 mm); normalization;

smoothing with a 8 mm FWHM kernel; denoising by linear regression of white matter and cerebrospinal fluid signals, realigning and scrubbing parameters; and finally linear detrending as well as band-pass filtering between 0.008 and 0.09 Hz. Again, there was no significant difference between groups in motion and we did not exclude any subject due to excessive motion. We entered the 11 ROIs that were constructed based on Dehghan et al. [8] as described above and calculated rsFC between each pair of ROI for each subject by averaging and correlating the time-series of each ROI.

### Statistical analysis

To explore differences in rsCBF, GMV and CoTh between patients and controls we performed generalized linear models based on whole-brain voxel-wise (vertex-wise for CoTh) analyses and ROI-analyses. The whole-brain analyses of rsCBF was the main statistical analysis. We considered disease status (fibromyalgia patients vs pain-free controls) as primary explanatory variable. When modelling the unadjusted group difference of rsCBF we included mean global rsCBF as co-variable. Similarly, we included total intracranial volume as co-variable when modelling unadjusted group differences of GMV and CoTh. To account for the effects of depression and anxiety we also modelled adjusted group differences of all neuroimaging markers and additionally included BDI-II and STAI-Trait t-value as co-variables. We followed this approach for both, whole-brain and ROI analyses of rsCBF, GMV and CoTh. We explored group differences in rsFC and calculated adjusted group differences of rsFC for each pair of the 11 pre-specified ROIs and compared rsFC between patients and controls for all resulting 55 connections. Again, we included BDI-ll and STAI-Trait t-values as co-variables to account for the effects of depression and anxiety. Age was corrected for in the study design using frequency matching of cases and controls according to age. To correct for multiple comparisons, we used family wise error (FWE) in all analyses withcorrection at peak-level in whole-brain analyses.

   We ran two sets of exploratory sensitivity analyses. First, we performed adjusted subgroup whole-brain analyses of the difference in rsCBF, GMV and CoTh between patients not taking any centrally acting drugs and their matched controls. Second, we explored the correlation of neuroimaging markers with clinical pain characteristics using Pearson correlation. We considered all neuroimaging markers with group differences at p≤0.10 in any of the analyses after correction for multiple comparisons. Due to the exploratory nature of these correlation analyses, multiple comparison correction was not employed. All reported p-values are two-sided and all confidence intervals refer to 95% boundaries. We performed statistical analyses for rsCBF and GMV in SPSS 23.0 (Version 23.0. Armonk, NY: IBM Corp., ROI-analyses, correlations)for CoTh in FreeSurfer (Version 5.3.0., http://surfer.nmr.mgh.harvard.edu/) and for rsFC in CONN (Version 15, http://www.nitrc.org/projects/conn).

## Results

### Study flow

We randomly selected and screened 151 fibromyalgia patients and 418 controls. The three most important reasons for excluding patients were neurologic or psychiatric co-morbidity other than unipolar depressive disorder (33 patients, 22%), inability to confirm fibromyalgia diagnosis at enrollment (23 patients, 15%) and inability to perform MRI (9 patients, 6%). The three most important reasons for excluding controls were neurologic or psychiatric co-morbidity (86 controls, 21%), chronic pain at enrollment (73 controls, 17%) and severe somatic co-morbidity (44 controls, 10%). We included 32 fibromyalgia patients and 32 age-matched

**Table 1. Baseline characteristics of fibromyalgia patients and pain-free controls.**

| | Fibromyalgia patients (N = 32) | Pain-free controls (N = 32) | p-value |
|---|---|---|---|
| **Socio-demographic characteristics** | | | |
| Age (years) | 50.7 (10.0) | 52.5 (11.2) | 0.49[+] |
| Higher education | 12 (38%) | 8 (25%) | 0.28˚ |
| Married | 20 (63%) | 19 (59%) | 0.80˚ |
| **Psychological characteristics** | | | |
| Depression (BDI-ll) | 20.7 (10.5) | 2.9 (4.3) | <0.001[+] |
| Anxiety (STAI Trait t-value) | 61.2 (9.1) | 46.8 (7.1) | <0.001[+] |
| **Pain characteristics** | | | |
| Average pain intensity during 24 hours prior to scanning (NRS) | 6.2 (1.9) | n.a. | n.a. |
| Pain intensity immediately before scanning (NRS) | 6.0 (2.2) | n.a. | n.a. |
| Degree of spread of pain (WPI) | 12 (4) | n.a. | n.a. |
| Disability (FIQ) | 61 (17) | n.a. | n.a. |

Values are numbers (percentage) or mean (standard deviation).

BDI-ll: Beck Depression Inventory Version 2

STAI: State Trait Anxiety Index

NRS: Numerical Rating Scale from 0 (no pain) to 10 (maximum pain)

WPI: Widespread pain index from 0 (no body region with pain) to 19 (all body regions with pain)

FIQ: Fibromyalgia Impact Questionnaire from 0 (no disability) to 100 (most severe disability)

[+]Students-t-test

˚Chi2 test

pain-free controls. All participants were female and right-handed. S1 Fig presents the study flow diagram.

## Socio-demographic, psychological and clinical characteristics

Table 1 presents the socio-demographic and psychological characteristics of the study population. Cases and controls were comparable in terms of age, education, and marital status. Patients had significantly higher scores for depression ($p \leq 0.001$) and anxiety ($p \leq 0.001$). Twenty-four patients (75%) had widespread body pain affecting more than 50% of their body and 20 patients (63%) reported average pain intensity of at least NRS 6. All patients took pain medications on a daily basis; 21 (66%) of them regularly took centrally acting drugs such as weak opioids, antidepressants or anticonvulsants (intake of strong opioids was an exclusion criterion).

## Differences in resting-state cerebral blood flow and correlation with pain characteristics

We found no increase in rsCBF in fibromyalgia patients as compared to pain-free controls in whole-brain analyses with and without depression and anxiety as additional co-variables. We found significant lower rsCBF in patients in the right Dorsolateral Prefrontal Cortex (x = 44; y = 32; z = 26; cluster size = 13 voxels; T = -4.39; $p_{FWE}$ = 0.03) in unadjusted analyses (Fig 1A) and in the left Inferior Middle Temporal Gyrus (x = -50; y = -50; z = -4; cluster size = 110 voxels; T = -6.01, $p_{FWE}$ = 0.002) in adjusted analyses (Fig 2A). However, rsCBF in these two regions was not correlated with average pain intensity 24h before or pain intensity immediately before neuroimaging, spread of pain or disability in 32 fibromyalgia patients (Figs 1 and 2B).

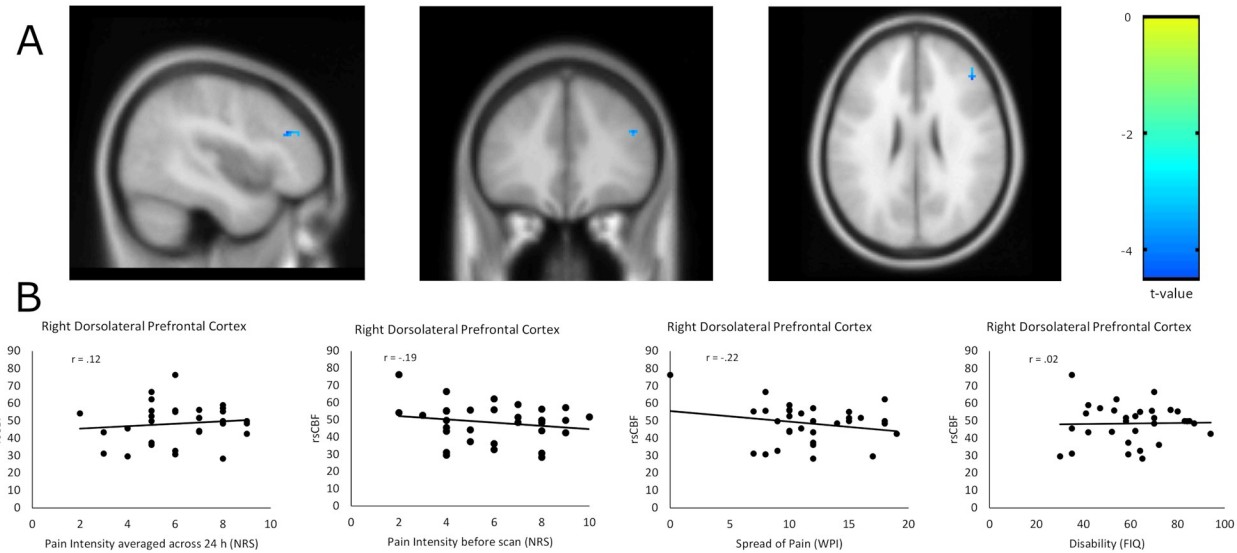

**Fig 1. Unadjusted difference in resting-state cerebral blood flow (rsCBF) between cases (N = 32) and controls (N = 32) in the right Dorsolateral Prefrontal Cortex (x = 44; y = 32; z = 26) controlling for mean global rsCBF at cluster-level of 10 voxels (A) and correlation of pain characteristics with rsCBF of this region within patients (N = 32) (B).** Displayed are cluster-sizes of >10 voxels with colors indicating t-values. NRS: Numerical Rating Scale from 0 (no pain) to 10 (maximum pain). WPI: Widespread pain index from 0 (no body region with pain) to 19 (all body regions with pain). FIQ: Fibromyalgia Impact Questionnaire from 0 (no disability) to 100 (most severe disability). r: correlation coefficients.

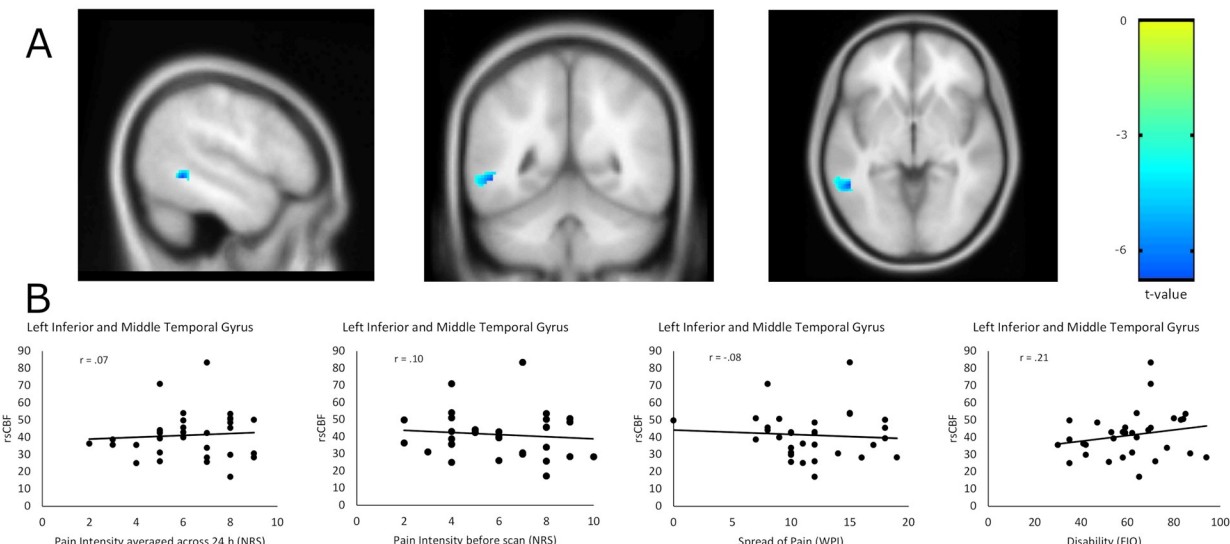

**Fig 2. Adjusted difference in resting-state cerebral blood flow (rsCBF) between cases (N = 32) and controls (N = 32) in the left Inferior Middle Temporal Gyrus (x = -50; y = -50; z = -4) controlling for mean global rsCBF, depression and anxiety at cluster-level of 10 voxels (A) and correlation of pain characteristics with rsCBF of this region within patients (N = 32) (B).** Displayed are cluster-sizes of >10 voxels with colors indicating t-values. NRS: Numerical Rating Scale from 0 (no pain) to 10 (maximum pain). WPI: Widespread pain index from 0 (no body region with pain) to 19 (all body regions with pain). FIQ: Fibromyalgia Impact Questionnaire from 0 (no disability) to 100 (most severe disability). r: correlation coefficients.

**Table 2. Adjusted differences in resting state perfusion (rsCBF, ml/100g/min) between 32 fibromyalgia patients and 32 pain-free controls in 11 pre-specified Regions of Interest.**

| MNI-coordinates | | | Brain area* | Mean rsCBF (SD) fibromyalgia patients | Mean rsCBF (SD) pain-free controls | Adjusted mean difference rsCBF (95% CI) | T-value | p-uncorr | P_FWE |
|---|---|---|---|---|---|---|---|---|---|
| x | y | z | | | | | | | |
| -48 | -27 | 24 | L insula | 36.65 (11.27) | 36.07 (11.12) | 1.97 (-5.98, 9.92) | 0.50 | 0.62 | 1.0 |
| 39 | 4 | 1 | R insula | 45.06 (13.63) | 43.43 (12.19) | 2.06 (-6.82, 10.93) | 0.46 | 0.65 | 1.0 |
| 43 | -2 | 1 | R insula | 45.74 (13.25) | 41.89 (14.70) | -1.57 (-11.46, 8.32) | -0.32 | 0.75 | 1.0 |
| -46 | -12 | 5 | L STG/insula | 52.60 (17.43) | 47.69 (14.60) | 1.24 (-10.34, 12.82) | 0.21 | 0.83 | 1.0 |
| 56 | -23 | 5 | R STG | 52.88 (12.50) | 50.46 (12.16) | 1.15 (-7.03, 9.33) | 0.28 | 0.78 | 1.0 |
| -15 | -48 | 72 | L SI | 25.52 (14.08) | 20.48 (45.91) | 10.15 (-16.50, 36.79) | 0.76 | 0.45 | 1.0 |
| -62 | -25 | 17 | L SII | 43.96 (11.01) | 42.49 (12.67) | 3.71 (-4.06, 11.47) | 0.96 | 0.34 | 1.0 |
| -2 | 48 | -12 | L ACC | 31.07 (19.80) | 35.97 (16.49) | -6.31 (-18.20, 5.59) | -1.06 | 0.29 | 1.0 |
| -17 | 30 | 31 | L MCC | 27.87 (10.07) | 27.39 (12.76) | -1.73 (-10.19, 6.74) | -0.41 | 0.68 | 1.0 |
| 13 | -55 | 7 | R lingual gyrus | 52.79 (13.08) | 47.92 (10.38) | 3.58 (-3.91, 11.07) | 0.96 | 0.34 | 1.0 |
| 27 | -12 | -15 | R amygdala | 47.79 (15.15) | 45.16 (13.26) | -4.14 (-14.63, 6.34) | -0.79 | 0.43 | 1.0 |

Results are data-based mean rsCBF with corresponding standard deviation (SD) and model-based adjusted mean differences of rsCBF with corresponding 95% confidence intervals (95% CI), t-values and p-values from multivariable general linear models.

* ROI selection based on Dehghan, M., et al., Coordinate-based (ALE) meta-analysis of brain activation in patients with fibromyalgia. Hum Brain Mapp, 2016. 37(5): p. 1749–58. with 27 voxels (216mm[3]) boxes centered on MNI-coordinates

Positive adjusted mean differences suggest increased regional rsCBF in patients as compared to controls after controlling for mean global rsCBF, depression and anxiety

MNI: Montreal Neurological Institute coordinates

FWE: Family Wise Error correction for multiple comparison L: Left, R: right

STG: superior temporal gyrus, SI: primary sensory cortex, SII: secondary sensory cortex, ACC: anterior cingulate cortex, MCC: middle cingulate cortex

## Differences in local resting state cerebral blood flow of 11 pre-specified ROIs

Table 2 shows adjusted group differences in local rsCBF of 11 pre-specified ROIs after controlling for mean global rsCBF, depression and anxiety. Although in more than half of the areas (7 of 11 ROIs) model-based adjusted mean difference between patients and controls suggest an increased blood flow in patients, none of the increases was statistically significant without nor with FWE correction. S1 Table shows unadjusted differences of rsCBF of ROI-analyses only including mean global rsCBF as co-variable. Again, we were unable to detect relevant group differences.

## Differences in resting-state functional connectivity, grey matter volume and cortical thickness

We found no differences in rsFC among areas associated with chronic pain in fibromyalgia in unadjusted or adjusted analyses after FWE correction. There was no evidence for differences in structural neuroimaging markers (GMV, CoTh) in patients as compared to controls in whole-brain analyses before or after FWE-correction. Tables 3 and 4 report group differences adjusted for total intracranial volume, depression and anxiety in GMV and CoTh of ROI-analyses. Patients showed non-significant decreases in structural neuroimaging markers in 8 of 10 ROIs for GMV and 12 of 23 ROIs for CoTh with and without FWE correction. S2 and S3 Tables showunadjusted differences of ROI-analyses of GMV and CoTh including only total intracranial volume as co-variable. As in adjusted analyses, we were unable to detect relevant group differences.

**Table 3. Adjusted differences in grey matter volume between (GMV) 32 fibromyalgia patients and 32 pain-free controls in 10 pre-specified Regions of Interest.**

| MNI coordinates | | | Brain area* | Mean GMV (SD) fibromyalgia patients | Mean GMV (SD) pain-free controls | Adjusted mean difference GMV (95% CI) | T-value | p-uncorr | P$_{FWE}$ |
|---|---|---|---|---|---|---|---|---|---|
| x | y | z | | | | | | | |
| -35 | 5 | 2 | L insula | 0.44 (0.05) | 0.45 (0.05) | -0.01 (-0.04, 0.02) | -0.63 | 0.53 | 1.0 |
| 39 | 5 | 1 | R insula | 0.43 (0.06) | 0.44 (0.05) | -0.01 (-0.04, 0.03) | -0.43 | 0.67 | 1.0 |
| -53 | -22 | 6 | L STG | 0.37 (0.04) | 0.38 (0.04) | -0.01 (-0.04, 0.02) | -0.56 | 0.58 | 1.0 |
| 58 | -23 | 5 | R STG | 0.35 (0.05) | 0.36 (0.04) | -0.01 (-0.04, 0.01) | -1.06 | 0.30 | 1.0 |
| -43 | -24 | 48 | L postcentral gyrus (SI) | 0.29 (0.03) | 0.30 (0.04) | -0.01 (-0.03, 0.02) | -0.58 | 0.56 | 1.0 |
| -47 | -10 | 13 | L rolandic operculum (SII) | 0.39 (0.05) | 0.40 (0.05) | -0.01 (-0.04, 0.02) | -0.59 | 0.56 | 1.0 |
| -4 | 34 | 13 | L ACC | 0.38 (0.05) | 0.39 (0.04) | 0.00 (-0.03, 0.03) | -0.20 | 0.84 | 1.0 |
| -6 | -16 | 40 | L MCC | 0.39 (0.05) | 0.40 (0.04) | 0.01 (-0.02, 0.03) | 0.43 | 0.67 | 1.0 |
| 16 | -68 | -5 | R lingual gyrus | 0.40 (0.04) | 0.42 (0.05) | -0.02 (-0.04, 0.01) | -1.26 | 0.21 | 1.0 |
| 27 | -1 | -19 | R amygdala | 0.51 (0.04) | 0.52 (0.04) | -0.02 (-0.04, 0.00) | -2.04 | 0.05 | 0.5 |

Results are data-based mean GMV with corresponding standard deviation (SD) and model-based adjusted mean differences of GMV with corresponding 95% confidence intervals (95% CI), t-values and p-values from multivariable general linear models.

* ROI selection based on Tzourio-Mazoyer, N., et al., Automated anatomical labeling of activations in SPM using a macroscopic anatomical parcellation of the MNI MRI single-subject brain. Neuroimage, 2002. 15(1): p. 273–89. Provided coordinates are the centers of mass of the ROIs

Negative adjusted mean differences suggest decreased GMV in patients as compared to controls after controlling for total intracranial volume, depression and anxiety

MNI: Montreal Neurological Institute coordinates. Coordinates are centres of mass.

FWE: Family Wise Error correction for multiple comparison

L: left, R: right

STG: superior temporal gyrus, SI: primary sensory cortex, SII: secondary sensory cortex, ACC: anterior cingulate cortex, MCC: middle cingulate cortex

### Subgroup analyses of patients not taking centrally acting drugs

Adjusted whole-brain subgroup-analyses, including depression, anxiety, and mean global rsCBF or total intracranial volume as co-variables, of 11 (34%) patients not taking centrally acting drugs and matched controls showed no group differences in rsCBF, GMV and CoTh.

## Discussion

### Main findings

To our knowledge, this is the first multimodal neuroimaging study integrating four different functional and structural markers of chronic pain in fibromyalgia. Our primary objective was to compare rsCBF patterns of fibromyalgia patients and pain-free controls using ASL as measure of neural activity at rest and thus likely to reproduce a neuroimaging marker of chronic, stimulus-independent pain. Based on the pathophysiological concept of enhanced central pain processes, we expected an increased neural activity at rest in pain processing areas, i.e., an increased rsCBF in these brain areas in cases as compared to controls. Contrary to our hypothesis, we found no evidence of increases in rsCBF in brain areas involved in pain processing in fibromyalgia neither in whole-brain nor in ROI analyses. Instead, we found decreased rsCBF in patients in the right Dorsolateral Prefrontal Cortex in whole-brain analysis after controlling for mean global rsCBF and in the left Inferior Middle Temporal Gyrus in whole-brain analyses adjusted for mean global rsCBF, depression and anxiety, even after correcting for multiple comparisons. However, rsCBF in these two areas did not correlate with pain characteristics such as pain intensity, spread of pain or disability in patients.

**Table 4. Adjusted differences in cortical thickness (CoTh, mm) between 32 fibromyalgia patients and 32 pain-free controls in 23 pre-specified Regions of Interest.**

| Region | Brain area* | Mean CoTh (SD) fibromyalgia patients | Mean CoTh (SD) pain-free controls | Adjusted mean difference CoTh (95% CI) | T-value | p-uncorr | P_FWE |
|---|---|---|---|---|---|---|---|
| L insula | L insula long G and S centralis | 3.15 (0.28) | 3.28 (0.29) | -0.18 (-0.40, 0.04) | -1.66 | 0.10 | 1.0 |
| | L insula short G | 3.56 (0.24) | 3.60 (0.27) | -0.06 (-0.25, 0.14) | -0.56 | 0.58 | 1.0 |
| | L insula anterior circular S | 2.72 (0.24) | 2.72 (0.23) | 0.12 (-0.06, 0.30) | 1.35 | 0.18 | 1.0 |
| | L insula inferior circular S | 2.69 (0.19) | 2.75 (0.19) | 0.00 (-0.14, 0.14) | -0.06 | 0.95 | 1.0 |
| | L insula superior circular S | 2.44 (0.17) | 2.43 (0.14) | 0.09 (-0.03, 0.21) | 1.48 | 0.14 | 1.0 |
| | R insula long G and S centralis | 3.35 (0.35) | 3.43 (0.29) | 0.01 (-0.23, 0.26) | 0.10 | 0.92 | 1.0 |
| | R insula short G | 3.46 (0.20) | 3.52 (0.20) | -0.05 (-0.21, 0.10) | -0.72 | 0.48 | 1.0 |
| | R insula anterior circular S | 2.69 (0.22) | 2.74 (0.26) | -0.02 (-0.20, 0.17) | -0.18 | 0.86 | 1.0 |
| | R insula inferior circular S | 2.67 (0.23) | 2.63 (0.19) | 0.10 (-0.07, 0.26) | 1.17 | 0.25 | 1.0 |
| | R insula superior circular S | 2.46 (0.14) | 2.45 (0.13) | 0.04 (-0.06, 0.15) | 0.81 | 0.42 | 1.0 |
| L STG | L transversal STG | 2.31 (0.21) | 2.36 (0.20) | -0.04 (-0.20, 0.12) | -0.50 | 0.62 | 1.0 |
| | L lateral STG | 2.98 (0.20) | 3.02 (0.22) | 0.03 (-0.13, 0.19) | 0.36 | 0.72 | 1.0 |
| | L STG planum polare | 3.39 (0.31) | 3.44 (0.25) | 0.08 (-0.12, 0.28) | 0.81 | 0.42 | 1.0 |
| | L STG planum temporale | 2.42 (0.26) | 2.52 (0.23) | -0.02 (-0.21, 0.17) | -0.20 | 0.84 | 1.0 |
| R STG | R transversal STG | 2.35 (0.24) | 2.43 (0.21) | -0.13 (-0.31, 0.04) | -1.53 | 0.13 | 1.0 |
| | R lateral STG | 2.98 (0.22) | 3.01 (0.22) | -0.02 (-0.19, 0.15) | -0.24 | 0.81 | 1.0 |
| | R STG planum polare | 3.28 (0.28) | 3.27 (0.25) | 0.06 (-0.13, 0.25) | 0.66 | 0.51 | 1.0 |
| | R STG planum temporale | 2.51 (0.29) | 2.49 (0.17) | -0.01 (-0.19, 0.18) | -0.10 | 0.92 | 1.0 |
| SI | L postcentral G | 2.17 (0.16) | 2.24 (0.18) | -0.05 (-0.18, 0.08) | -0.73 | 0.47 | 1.0 |
| SII | L subcentral G and S | 2.60 (0.15) | 2.65 (0.19) | -0.04 (-0.17, 0.09) | -0.63 | 0.53 | 1.0 |
| L Cingulate Cortex | L ACC | 2.53 (0.17) | 2.57 (0.22) | 0.01 (-0.14, 0.15) | 0.08 | 0.94 | 1.0 |
| | L MCC | 2.57 (0.18) | 2.58 (0.22) | 0.11 (-0.04, 0.26) | 1.49 | 0.14 | 1.0 |
| | R lingual G | 2.01 (0.13) | 2.02 (0.13) | -0.05 (-0.15, 0.05) | -1.01 | 0.31 | 1.0 |

Results are data-based mean CoTh with corresponding standard deviation (SD) and model-based mean differences of CoTh with corresponding 95% confidence intervals (CI), t-values and p-values from multivariable general linear models.

* ROI selection based on Destrieux, C., et al., Automatic parcellation of human cortical gyri and sulci using standard anatomical nomenclature. Neuroimage, 2010. 53 (1): p. 1–15.) No coordinates are provided as this is a surface-based analysis.

Negative adjusted mean differences suggest decreased CoTh in patients as compared to controls after controlling for total intracranial volume, depression and anxiety

FWE: Family Wise Error correction for multiple comparison

L: left; R: right, G: gyrus, S: sulcus

STG: superior temporal gyrus, SI: primary somatosensory cortex, SII: secondary somatosensory cortex, ACC: anterior cingulate cortex, MCC: middle cingulate cortex.

We believe that the decrease in rsCBF in the Dorsolateral Prefrontal Cortex is likely to be an effect of depression as the role of the Dorsolateral Prefrontal Cortex in depression is well established [42, 43] and the effect could not be observed anymore after controlling the analysis for depression. With regard to the decreased rsCBF in the Inferior Middle Temporal Gyrus we propose that this effect may either indicate a process secondary to the pain experience, such as the subjective evaluation of pain, or a false positive statistical finding rather than an effect

attributable to pain. The rationale behind this is fourfold. First, there is no prior evidence for a pathophysiological involvement of the Inferior Middle Temporal Gyrus in fibromyalgia. Second, we detected a decreased rsCBF in this area in patients as compared to controls which is contrary to our main hypothesis of increased rsCBF in cases as result of central hypersensitivity. Third, we did not find any relevant correlations between rsCBF in this area and pain characteristics. Fourth, we were unable to see any group differences between cases and controls in this area for other neuroimaging marker. In view of the large number of analyses conducted, a false positive statistical finding is a possible explanation."

Our secondary objective was to compare resting-state functional connectivity between pre-specified areas likely to be involved in pain processing, grey matter volume and cortical thickness between cases and controls assuming a reduction in these neuroimaging markers in fibromyalgia. Again contrary to our hypotheses, we neither found evidence of decreased rsFC, GMV or CoTh in brain areas involved in pain processing of fibromyalgia patients. The results remained robust in sensitivity analyses comparing fibromyalgia patients not taking centrally acting drugs with corresponding controls.

### Scientific context of our findings

There is ongoing debate to what extent brain areas processing acute pain signals are also involved in the development and maintenance of chronic, stimulus-independent pain [8, 17]. We therefore defined the whole-brain analysis as the main statistical approach and performed secondary ROI analyses based on the recent meta-analysis by Dehghan and colleagues as most comprehensive evidence synthesis of functional and structural alterations in the central nervous system of fibromyalgia patients [8]. Previous neuroimaging studies typically focused on the comparison of acute pain processing mechanisms between fibromyalgia patients and pain-free controls using experimentally-evoked pain paradigms [7]. However, neuroimaging of chronic, stimulus-independent pain requires a different approach. Chronic pain typically remains constant during the course of an imaging session, rendering it invisible to traditional imaging techniques using pain paradigms. Task-free, resting-state parameters such as rsCBF are markers for brain activity at rest and thus more appropriate to measure chronic pain [18–20]. ASL is the sequence of choice to measure rsCBF [17]. Still, studies using rsCBF based on ASL are sparse in pain research with only one study up until now performed in fibromyalgia [24–29]. Although these studies employed a resting-state paradigm, none them integrated functional and structural neuroimaging by using multimodal scanning methods and none of them considered the effect of centrally acting drugs typically used by chronic pain patients on their results. Furthermore, only some of these studies addressed important confounders such as age, gender and concomitant depression. To our knowledge, Shokouhi and colleagues conducted the only study using ASL in fibromyalgia investigating resting state perfusion patterns [29]. They compared 23 patients with 16 pain-free controls and thus included a much smaller and unmatched study population as compared to our study. They reported hypoperfusion in the putamen in subjects with chronic pain that was correlated with degree in disability but not pain intensity thus suggesting that this hypoperfusion was due to adaptation processes. The correlation between rsCBF in the Putamen and disability was positive when using the Pain Disability Index to measure disability but negative when using the Fibromyalgia Impact Questionnaire as other measure of disability. The authors didn't comment on this conflicting finding. However, they didn't find group differences of rsCBF between fibromyalgia patients and controls what is in line with our findings. With regards to the literature on functional connectivity in fibromyalgia patients, our objective was to compare rsFC between pre-specified areas likely to be involved in pain processing as defined by Dehghan and colleagues [8]. Previous

studies investigated rsFC of the Default Mode Network. These studies found an increased connectivity within the Default Mode Network and between the Insula and the Default Mode Network [21, 22]. The most recent meta-analysis investigating GMV showed grey matter decreases in the anterior and posterior cingulate cortex, the medial prefrontal cortex and the parahippocampal gyrus or fusiform cortex [44]. However, the original studies which were included in this meta-analysis [9–13] and the latest study investigating CoTh [14] typically did not correct for co-morbid depression or for multiple comparisons, and if they did, group differences in neuroimaging markers lost statistical significance [9–14]. Therefore, our inability to find relevant group differences in structural neuroimaging markers are in line with the findings of these previous studies [9–14]. Furthermore, evidence generation in the field of neuroimaging and fibromyalgia is still in its early stage. This phase of evidence generation is often characterized by conflicting findings as expression of the evidence accumulating process itself, previously referred to as proteus phenomenon [45, 46]. Therefore, our inability to find evidence to support the hypothesis of altered central pain processing might also be interpreted in the light of this phenomenon.

## Strengths and limitations

Major strengths of the present study include integration of four different functional and structural neuroimaging markers using a multimodal scanning protocol, evaluation of rsCBF using ASL; adjustment for depression and anxiety; correction for multiple comparisons and performance of a sensitivity analysis in a subgroup of patients not taking centrally acting drugs. Although fibromyalgia is characterized by intensive, widespread pain not sufficiently explained by peripheral lesions [7, 12] making it a good model to study the pathophysiology of enhanced central pain mechanisms it also shows high co-morbidity with depression and anxiety. This may bias neuroimaging of chronic pain since brain areas involved in pain processing overlap with those involved in the pathophysiology of depression, e.g. prefrontal, cingulate, and supplementary motor cortex [39]. Pain processing is tightly linked to top-down emotional control processes involving prefrontal cortices and amygdala [47]. Hence, we adjusted all our analyses for depression and anxiety and again were unable to find any group differences. Another problem of studies in fibromyalgia is the long-term treatment with centrally acting drugs, which may also interfere with neuroimaging markers. For possible rebound-effects and ethical reasons we decided not stop current medication. To address this limitation, we performed a subgroup-analysis including only patients not taking any centrally acting drugs and matched controls. We again found no group differences in whole-brain analyses of rsCBF, GMV and CoTh. We recruited both cases and controls in the only tertiary care hospital in the capital of Switzerland, with the same referral pathways for fibromyalgia patients and pain-free controls. This allowed us sampling cases and controls from the same source population, which is important to avoid selection bias in case-control studies. Severity of fibromyalgia measures by the FIQ in our patients was similar to patients included in key randomized controlled trials evaluating non-pharmacological interventions [48, 49] and phase III effectiveness trials of different drugs [50, 51]. Our study included totally 64 participants, making it one of the largest in the field. Therefore, we do not think that our inability to observe statistically significant group differences was merely due to a lack of power. Our results were consistent across four imaging modalities and robust to the type of analysis (whole-brain and ROI-analyses, subgroup analysis in patients not taking centrally acting drugs). Additionally, we were unable to detect relevant correlation between neuroimaging markers with clinically important outcomes such as pain intensity, spread of pain and disability. Still, an argument for limited power to detect relevant group differences is the fact that patients showed numerically increased rsCBF, decreased

GMV and decreased CoTh in most of the pre-specified ROIs even after adjusting for depression and anxiety what would be in line with our a priori hypotheses. Previous studies showing group-differences were both smaller and of less methodological rigor as compared to our study. They typically did not account for co-morbid depression and intake of pain-modulating, centrally acting drugs even though both features are particularly frequent in fibromyalgia and likely to influence neuroimaging results. Furthermore, by nature, case-control studies tend to inflate group-differences because severely sick cases are compared to healthy controls. This suggests that we would have detected moderate differences if they had been present. However, we did not acquire any fieldmaps and therefore could not correct our analyses for magnetic field inhomogeneity potentially leading to decreased sensitivity in detecting group differences. Yet, we believe that correcting our analyses for field inhomogeneity would not have changed the interpretation of our findings. Signal quality would have improved in both groups, the cases and controls alike and a more sensitive method would only have detected small group differences. The relevance of these small effect sizes, if at all present, would be equivocal.

## Implications

Our findings do not necessarily imply that altered central pain processing is not involved in fibromyalgia but may demonstrate that currently available functional and structural neuroimaging markers are not able to map stimulus-independent pain. Stimulus-independent, chronic pain may involve subtle functional and structural alterations which might be missed in medium-sized case-control studies like the present one and could be detected by larger multimodal studies also using resting-state designs. Because we analyzed brain areas involved in the processing of acute pain signals, our results suggest that these areas are unlikely to be involved in the development and maintenance of chronic, stimulus-independent pain.

## Supporting information

**S1 Fig. Flow diagram of fibromyalgia patients and controls clinically evaluated between 1st July 2011 and 30th June 2013 and recruited for the study between 1st November 2013 and 31st January 2015 at the University Hospital of Bern.** $ according to criteria of American College of Rheumatology; *patients with light opioids, antidepressants, pregabalin or gabapentin included; ˚5 patients with mental retardation, 3 patients with pregnancy.
(DOCX)

**S1 Table. Unadjusted differences in resting state perfusion (rsCBF, ml/100g/min) between 32 fibromyalgia patients and 32 pain-free controls in 11 pre-specified Regions of Interest.** Results are data-based mean rsCBF with corresponding standard deviation (SD) and model-based unadjusted mean differences of rsCBF with corresponding 95% confidence intervals (CI), t-values and p-values from multivariable general linear models.
(DOCX)

**S2 Table. Unadjusted differences in grey matter volume between (GMV) 32 fibromyalgia patients and 32 pain-free controls in 10 pre-specified Regions of Interest.** Results are data-based mean GMV with corresponding standard deviation (SD) and model-based unadjusted mean differences of GMV with corresponding 95% confidence intervals (CI), t-values and p-values from multivariable general linear models.
(DOCX)

**S3 Table. Unadjusted differences in cortical thickness (CoTh, mm) between 32 fibromyalgia patients and 32 pain-free controls in 20 pre-specified Regions of Interest.** Results are

data-based mean CoTh with corresponding standard deviation (SD) and model-based unadjusted mean differences of CoTh with corresponding 95% confidence intervals (CI), t-values and p-values from multivariable general linear models.
(DOCX)

## Acknowledgments

We would like to thank Katrin Ziegler, MSc, Clinical Trials Unit, University of Bern, Switzerland, for her support in data-management and Fabienne Treichel, study nurse, for her support in patient recruitment and data collection.

## Author Contributions

**Conceptualization:** Monika Müller, Roland Wiest, Niklaus Egloff, Stephan Reichenbach, Aristomenis Exadaktylos, Peter Jüni, Michele Curatolo, Sebastian Walther.

**Data curation:** Monika Müller, Florian Wüthrich, Andrea Federspiel, Roland Wiest.

**Formal analysis:** Florian Wüthrich.

**Funding acquisition:** Monika Müller, Michele Curatolo, Sebastian Walther.

**Investigation:** Monika Müller, Florian Wüthrich, Andrea Federspiel, Niklaus Egloff, Stephan Reichenbach, Aristomenis Exadaktylos.

**Methodology:** Monika Müller, Florian Wüthrich, Roland Wiest, Peter Jüni, Michele Curatolo, Sebastian Walther.

**Project administration:** Michele Curatolo, Sebastian Walther.

**Supervision:** Michele Curatolo, Sebastian Walther.

**Validation:** Florian Wüthrich.

**Writing – original draft:** Monika Müller.

**Writing – review & editing:** Florian Wüthrich, Andrea Federspiel, Roland Wiest, Niklaus Egloff, Stephan Reichenbach, Aristomenis Exadaktylos, Peter Jüni, Michele Curatolo, Sebastian Walther.

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
