## [Decision Letter · Decision Letter 0]

6 Oct 2020

PONE-D-20-19081

Altered central pain processing in fibromyalgia – a multimodal neuroimaging case-control study using arterial spin labelling

PLOS ONE

Dear Dr. Curatolo,

Thank you for submitting your manuscript to PLOS ONE. After careful consideration, we feel that it has merit but does not fully meet PLOS ONE’s publication criteria as it currently stands. Therefore, we invite you to submit a revised version of the manuscript that addresses the points raised during the review process.

We look forward to receiving your revised manuscript.

Kind regards,

Claudia Sommer

Academic Editor

PLOS ONE

Journal Requirements:

Reviewers' comments:

Reviewer's Responses to Questions

**Comments to the Author**

1. Is the manuscript technically sound, and do the data support the conclusions?

Reviewer #1: Yes

Reviewer #2: Yes

2. Has the statistical analysis been performed appropriately and rigorously? 

Reviewer #1: Yes

Reviewer #2: Yes

3. Have the authors made all data underlying the findings in their manuscript fully available?

Reviewer #1: Yes

Reviewer #2: Yes

4. Is the manuscript presented in an intelligible fashion and written in standard English?

Reviewer #1: Yes

Reviewer #2: Yes

5. Review Comments to the Author

Reviewer #1: Dear Prof. Sommer,

in the article "Altered central pain processing in fibromyalgia - a multimodal neuroimaging case-control study using arterial spin labeling" the authors use classical T1 MRI imaging, arterial spin labeling (ASL) and functional BOLD brain connectivity analysis to compare fibromyalgia patients with healthy controls without a specific stimulus paradigm. They also use questionnaires to investigate the severity of the core symptoms of fibromyalgia (FMS).

All investigations have already been published on FMS patients and controls. There are not so many publications on ASL, but there are a lot on rs-fMRI and structural cortex measurements. A positive feature of the study is that they publish null findings, which is in contrast to the usual publications in the structural and connectivity area. In contrast to these studies, their number of subjects (32 per group) is relatively high (upper average) and the authors correct their statistical results, except for the correlation analyses, by multiple comparisons (FWE or FDR). Furthermore, in their ROI analyses they selected the ROIs by a-priori hypotheses of a previous meta-analysis. The methods used are in accordance with current standards in neuroimaging and support the statements in the results section.

1. 1. 71 and elsewhere: These quotations are from older publications, there are now more recent reviews and publications.

2. 2. 81: "as an advanced" + 82 "a direct marker"

3. 3. 105: "confirmed with the diagnosis"

4. 4. 110: An explanation why unipolar depressions were included would be useful here. Furthermore, it should be described in more detail what is meant by end-stage somatic comorbidity, cardiovascular diseases can also have an influence on functional and structural brain markers.

5. 5. 166: Was this preprocessing also done with freesurfer? Structural preprocessing and cortex analysis as well as GMD are mixed up in this paragraph, easier to understand would be to summarize all structural analyses.

6. 6. 187: Here it should be justified why no fieldmaps were taken up to be able to calculate magnetic field inhomogeneities.

7. 7. 189: Here it is unclear whether the classical CONN-ROIs were used or extra ROIs of the Deghan et al publication were implemented.

8. 8. 205: Were these clinical parameters included in the model as statistical co-variables, or should "adjusted" be defined as such?

9. 9: The correct citations to the analytical programs should be added in the methodological paragraph.

10. 10. 241: Were these drugs also taken on the day of the MRI examination?

11. Table 1: Are all results distributed normally, so the mean value is appropriate?

12. 12. 255: The DLPFC does not seem to be in the Dehghan regions, in this paragraph a clearer distinction should be made between ROI and whole brain analysis.

13. 13. 261: According to the table there are 10 regions which have a lower mean regional blood flow in fibromyalgia patients, is the mean difference meant here?

14. 14. 297: Up to this paragraph it is still not clear to the reader what is meant by "crude" or adjusted analyses. Are the drug subgroups referred to here, or the clinical parameters?

15. 15 Table 3: In which unit or on which scale is GMD measured? Same question for Table 4 and cortical thickness.

16. 16. 373: Depressive symptoms are, however, one of the core symptoms of FMS patients, if the authors only want to measure chronic pain in the brain, this hypothesis is correct, for the "fibromyalgia signature" one would have to exclude major depressions and then analyse them again.

17. 17. 382: Before the authors give this statement, they should consult Üçeyler et al 2013 and the many following publications on functional and structural loss from the small nerve fibres in some FMS patients.

18. 18. 392: How large was the proportion of patients without medication?19. 405: The authors could check how severely their patients were affected compared to the rest of the worldwide patient population by comparing them with other publications using the FIQ or the NRS scale.

19. 20. 406: In the whole discussion there are no studies mentioned, which for example see clear changes in functional connectivity, even with multiple comparisons. The same applies to the structural parameters. Even if other results were found, other studies with the same methods should be mentioned.

20. 21. 415: This statement cannot be supported by the results, because no chronic pain processing regions were found. Rather, as mentioned before, the methodology is not sensitive enough.

21. 22 Figure 1: It should be clear in which unit the Colour Bar displays, the same applies to Figure 2.

Overall, however, the results and methods support the statements of the authors of this study. Thank you for the opportunity of the review.

Reviewer #2: The manuscript “Altered central pain processing …” reports on a multimodal MRI study of 32 fibromyalgia patients compared to 32 matched controls. The investigators analyzed resting state cerebral blood flow by arterial spin labeling, resting state functional connectivity, grey matter density, and cortical thickness. The investigators adjusted their analyses to consider multiple different aspects of their participants, including demographics, mood, and the use of pharmaceutics. The emphasis of the paper is on the novel use of arterial spin labeling, which did not turn out as expected, but what is striking to the reviewer is that, despite careful analysis, the investigators did not find any of the differences that have been reported in the literature for gray matter density, etc.

Reviewing the methods, the case ascertainment for fibromyalgia patients seems reasonable, using the 2011 criteria. Measures of pain and mood use appropriate instruments. The 12-channel head coil seems like an older coil compared to the newer 64-channel coils, but such should only affect spatial resolution but not impact the ability to detect altered functional measures. The methods appear generally sound with a substantial sample size that should be able to detect changes in each of the measurements made.

To follow are comments about ways to clarify the methods further:

• Lines 154-7: Make it clear what the 11 ROIs are. It isn’t obvious from the list here that there are 11 ROIs. Should inform the reader that there are multiple insula ROIs (one ROI in left insula, and two in right insula) defined by Dehghan.

• S1 Table: Is the 40 for number of ROIs in the header of S1 Table a mistake? The results seem to show only the same ROIs as for the other analyses. Also, these 40 ROIs are not mentioned in the section entitled “Selection of Region of Interests (ROIs)”.

• Line 180-1: Unclear what is meant by “We finally modelled a mean rsCBF map for each subject based on 40 pre-processed maps…”. Explain what the 40 pre-processed maps are and where they came from. We assume these are for each of the selected ROIs.

• Lines 207-8: It is unclear why FWE correction was used for whole brain analyses at peak rsCBF and for cortical density, but FDR correction was used for ROI analyses. A consistent method for correction for multiple comparisons should be used, otherwise, a justification should be given.

• Lines 208-9: A cluster size threshold of 10 voxels is not justified.

o However, this is very low threshold and would bias the results away from the null. As the results were the null, this does not seem to be problematic in regards to the results.

• Lines 215-6 state that “We considered all neuroimaging markers with group differences at p ≤ 0.10 in any of the crude or adjusted analyses after correction for multiple comparisons for these correlations.” I believe “for these correlations” should be removed here as it seems you are discussing group differences in imaging results, not correlations. Also, your p value is .10 corrected for multiple comparisons, but the tables show uncorrected p values. Explain why this was done or correct this.

• Line 216: Using a threshold of .7 for significant correlations seems high. Is there justification for using .7?

o However, this permissive number again would bias away from the null – which does not seem to be impacting the analysis here.

• Define what is meant by “crude”. I suggest using a term other than “crude” as “crude” has negative connotations. Lines 259-60 should state for what the analysis for Table 2 was adjusted. Line 262 should state for what S1 Table was adjusted. Also, line 94 states that, “We adjusted all our analyses for depression and anxiety…” (similarly, lines 205-6), but it is not clear what a “crude” analysis is. Line 210 restates this.

• Specify what variable has been adjusted for rsCBF of grey matter in S1 Table.

• Tables 2, 3, 4, and S1: The rsCBF mean differences are different from actual numerical differences, but it is not clear why.

• S1 Table: Were the mean differences here adjusted or not? It should say in the header, similar to Table 2.

• Line 306 (Table 3): What are “adjusted mean differences” here? If they are adjusted for total intracranial volume, depression and anxiety, state that in the header or text. Are the mean GMDs for each patient group adjusted also? Headers for Tables 2 and 3 should state that a 3x3x3 voxel box around the MNI coordinates is used as the ROI, not the full brain region. It would be preferable for the mean of the full ROI be used, rather than just the mean of a small box.

Overall, the authors do a very credible job in finding no differences between the groups. These negative results are very important to report – as previous reports of changes in these imaging parameters are frequently used as evidence of the ‘pathology’ of fibromyalgia. Demonstrating that evidence of altered neurological activity at resting that would account for differences in pain perception cannot be readily demonstrated is also one of several observations that undermine the concept of “central sensitization” being the cause of fibromyalgia. These results are important for the field to be aware of.

The greatest weakness of this study is that no measurements of pain perception were made during the scanning sessions. While scores estimating the burden of pain experienced within the last 24 hours before the scan were collected, the investigators cannot state with certainty that the fibromyalgia patients were experiencing clinically relevant pain while MRI measures were made. It also seems that strongest correlative work would relate the sensation occurring at the time of scanning to the scans.

In regards to the discussion, I would opine it can be approved by addressing the comments below:

• What do their ASL findings mean? The authors do note decreased rsCBF in th r dorsolateral PFC and l inferior middle TMG that do not correlate with pain measures. Can these changes be part of fibromyalgia pathology – or do they reflect false positive statistical phenomenon related to performing multiple comparison analyses? Please comment.

• This is the largest study to date to look at rsfunctional connectivity, gray matter volume, and cortical thickness – and perhaps the first to show negative findings this definitively. These negative findings only receive two sentences in the discussion (lines 372-375). Considering that these types of changes reported in smaller and poorly adjusted papers are often cited as evidence of the pathology of fibromyalgia, I opine that the authors should comment on this in the discussion.

• Add lack of in-MRI pain measurements to the strength and limitations section.

• The authors may want to comment on the proteus phenomenon (https://www.ncbi.nlm.nih.gov/pmc/articles/PMC1182327/) in their discussion about false negatives (lines 400-406).

• Line 409: ‘may point out’ should be ‘demonstrate’.

Here are some minor comments to consider as well:

Minor comments:

- I recommend not using “CT” as the abbreviation for “cortical thickness”, since it is more often used to mean “CAT scan”.

- Line 42: Change “as brain” to “as brain areas” or “as regions of interest”.

- Line 165: place “MNI” in parentheses. Use “smoothed” instead of “smoothened”.

- Line 171: “TI2” is defined in the text but it is not used in the equation, so remove it.

- Lines 229, 231: Change “enrolment” to “enrollment”.

- Table 1: Don’t use “*” to mean Student’s T test as it is commonly used to signify statistical significance.

- Line 256 and elsewhere: Use “voxels” rather than “voxel” to specify the cluster sizes.

- Line 261-2: I suggest changing to wording of “None of the increases was statistically significant neither without nor with FDR correction” to “None of the increases was statistically significant with or without FDR correction”.

- Line 265: Figure 1 text: Clean up the wording so as not to repeat descriptions of A) and B). Same for Figure 2 text.

- Line 282: Change “WIP” to “WPI”.

- Line 288: Remove the first instance of “Dehghan et al.” as it doesn’t make sense to have the citation immediately followed by the full reference.

- Line 294: Move the definition of SI to before the definition of SII.

- Line 297: I suggest changing “We found no differences in rsFC among areas associated with chronic pain in fibromyalgia neither in crude not adjusted analyses after FDR correction” to “We found no differences in rsFC among areas associated with chronic pain in fibromyalgia in adjusted or unadjusted analyses after FDR correction”.

- Line 300: Remove the dash in “group-differences”. Also on line 328.

- Line 305: Place “(GMD)” immediately after “grey matter density”. Also, use either “pre-defined” or “pre-specified” or “a priori” consistently when referring to the ROIs you used, not all of them.

- Lines 312 and 322: FDR is not used in the Tables so remove it from the footers for Tables 3 and 4 (or add FDR corrected results).

- Line 314: Add definition of SI.

- Line 327: State for what the subgroup’s analyses were adjusted.

- Lines 355-6: Clarify that pain may be invisible at the individual level, but not at the group level when comparing patients to healthy controls.

- Line 371: add “and controls” to the end of “they did not find group differences of rsCBF between fibromyalgia patients”.

- Line 387 states that you found “robust” results, but this statement does not seem supported as most of your results were not significant.

6. PLOS authors have the option to publish the peer review history of their article (what does this mean?). If published, this will include your full peer review and any attached files.

Reviewer #1: **Yes: **Hans-Christoph Aster

Reviewer #2: **Yes: **Brian Walitt

---

## [Decision Letter · Decision Letter 1]

19 Jan 2021

Altered central pain processing in fibromyalgia – a multimodal neuroimaging case-control study using arterial spin labelling

PONE-D-20-19081R1

Dear Dr. Curatolo,

We’re pleased to inform you that your manuscript has been judged scientifically suitable for publication and will be formally accepted for publication once it meets all outstanding technical requirements.

Kind regards,

Claudia Sommer

Academic Editor

PLOS ONE

Additional Editor Comments (optional):

Reviewers' comments:

Reviewer's Responses to Questions

**Comments to the Author**

1. If the authors have adequately addressed your comments raised in a previous round of review and you feel that this manuscript is now acceptable for publication, you may indicate that here to bypass the “Comments to the Author” section, enter your conflict of interest statement in the “Confidential to Editor” section, and submit your "Accept" recommendation.

Reviewer #1: All comments have been addressed

Reviewer #2: All comments have been addressed

2. Is the manuscript technically sound, and do the data support the conclusions?

Reviewer #1: Yes

Reviewer #2: Yes

3. Has the statistical analysis been performed appropriately and rigorously? 

Reviewer #1: Yes

Reviewer #2: Yes

4. Have the authors made all data underlying the findings in their manuscript fully available?

Reviewer #1: Yes

Reviewer #2: Yes

5. Is the manuscript presented in an intelligible fashion and written in standard English?

Reviewer #1: Yes

Reviewer #2: Yes

6. Review Comments to the Author

Reviewer #1: (No Response)

Reviewer #2: The authors have addressed all issues from the prior review in a satisfactory manner to this reviewer.

7. PLOS authors have the option to publish the peer review history of their article (what does this mean?). If published, this will include your full peer review and any attached files.

Reviewer #1: **Yes: **Hans-Christoph Aster

Reviewer #2: **Yes: **Brian Walitt

---

## [Editor Report · Acceptance letter]

22 Jan 2021

PONE-D-20-19081R1 

Altered central pain processing in fibromyalgia – a multimodal neuroimaging case-control study using arterial spin labelling 

Dear Dr. Curatolo:

I'm pleased to inform you that your manuscript has been deemed suitable for publication in PLOS ONE. Congratulations! Your manuscript is now with our production department. 

Kind regards, 

on behalf of

Prof. Dr. Claudia Sommer 

Academic Editor

PLOS ONE